# Study on the Design and Mechanical Properties of a Novel Hexagonal Cell Body Topology

**DOI:** 10.3390/polym16152201

**Published:** 2024-08-02

**Authors:** Enze Hao, Xindan Zhang, Xueqing Zhao, Hui Zhang

**Affiliations:** 1School of Mechanical Engineering and Automation, Dalian Polytechnic University, Dalian 116034, China; hez_0809@163.com (E.H.); zxd_jx2014@163.com (X.Z.); zhaoxueqing99@163.com (X.Z.); 2Dalian Jinshiwan Laboratory, Dalian 116034, China

**Keywords:** bio-inspired, honeycomb topology, fused deposition modeling, mechanical properties

## Abstract

The honeycomb structure is a topological structure with excellent performance that stems from the properties of the basic units of the structure. Different structural features of basic units may lead to different mechanical characteristics in the whole part. In this study, a novel hexagonal cell body topology structure (NH) was designed and manufactured by the fused deposition modeling (FDM) technique to explore the effects on mechanical properties. The tensile and impact performance of the NH structure were compared with the regular hexagonal honeycomb structure (HH), and the influence of different unit single-cell sizes on the impact performance of the NH structure was investigated. The force transmission of the basic units of the NH structure was revealed through finite element analysis. The results indicate that both the tensile and impact performances of the NH structure have been improved compared to the HH structure. The improvement is due to the better force transmission capability of the basic units of the NH structure, leading to a more uniform stress distribution. Moreover, excessively large or small single-cell sizes of the NH structure will reduce the overall structure’s impact resistance. The overall structure achieves optimal impact resistance when the single-cell size is around 1.2 mm.

## 1. Introduction

Biomimetic structures, especially honeycomb structures, have been rapidly and widely used in medical biology [1], construction engineering [2], aerospace [3,4], automotive engineering [5], and other fields, with the development of 3D printing technology. The honeycomb structure is a kind of topological structure with a two-dimensional cell array stacked parallel inside and outside the plane [6], making the overall structure produce higher porosity and lower density compared with the matrix material and obtain higher specific strength and specific energy absorption [7]. In addition, reasonable design of the structure of the microscopic cells can give the honeycomb structure other properties, such as a negative Poisson’s ratio [8], negative thermal expansion [9], and negative stiffness [10].

In 2001, Hales [11] first proved the hexagonal honeycomb structure conjecture that bees build hives that consume the least amount of material but provide the most space. This proves that hexagonal cell configuration is the most efficient structure in nature, so people begin to conduct in-depth research on hexagonal honeycomb structures. Subsequently, according to different cell shapes, triangular, square, and circular cells were proposed, and the above four kinds of cells have been studied through experimental, theoretical, and numerical analysis methods [12,13,14]. For example, Bezazi et al. [15] calculated Poisson’s ratio and Young’s modulus of a center symmetric honeycomb structure under a uniaxial load through finite element analysis and found that the stiffness and other characteristics of the finished structure could be affected by modifying the bottom wall thickness ratio. Du et al. [16] manufactured a new layered thermoplastic composite honeycomb cylinder structure and studied its axial static compression performance. The results showed that the laying mode of different coaxial bars had an impact on the compression performance of the structure, and the staggered laying mode was better than the regular laying mode.

In addition to the above unit configuration, the re-entrant honeycomb structure can be formed by concaving any two opposite vertices of the traditional hexagonal convex honeycomb structure inward. The re-entrant honeycomb structure is a typical negative Poisson ratio material with typical “stretch” characteristics [17]. In recent decades, the types of honeycomb structures with negative Poisson’s ratio characteristics have been expanded, and metamaterials such as chiral honeycomb, star-shaped honeycomb, and double-V-shaped honeycomb have successively emerged. For example, Liu et al. [18] combined cellular structures and chiral structures to form a new reentrant structure, which is compared with the biomimetic scapular re-entry structure and double-crank re-entry structure. The average carrying capacity is increased by at least 55% and 13.2%, respectively, and the specific energy absorption is increased by at least 8.6% and 3.4%, respectively. Li et al. [19] applied the negative Poisson’s ratio structure to the acceleration measurement study, converted the change of transmission of the negative Poisson’s ratio honeycomb structure caused by elastic deformation under acceleration into the change of solar cell output, and verified the effectiveness of the proposed method through geometric analysis and experiments. Gao et al. [20] developed a finite element model of a three-dimensional double-v chiral honeycomb structure, fabricated a physical object for quasi-static compression tests, and proposed a method for Poisson’s ratio and Young’s modulus prediction considering the influence of neighboring cells, which proved the credibility of the method. Buckmann et al. [21] designed a novel three-dimensional re-entrant auxiliary structure, which can realize a special structure with a positive/zero/negative Poisson’s ratio through different assembly methods.

Existing research has achieved some results in the study of cellular configuration and property requirements. Combined with extrusion 3D printing, such as fused deposition manufacturing (FDM), this paper provides a novel hexagonal cell body topology suitable for extrusion 3D printing by modifying the cells of regular hexagonal honeycomb structure. The tensile and impact properties of the new structure were studied by means of experiment and simulation. Compared with the regular hexagonal honeycomb structure (HH structure), the mechanical properties of the new structure were verified. In addition, the effects of different single-cell sizes on the impact properties of the new structures were studied. Finally, the stress of the new structure was discussed by using the finite element analysis method, and the mechanism of mechanical property enhancement of the new structure was revealed.

## 2. Materials and Methods

### 2.1. Design of the Novel Hexagonal Cell Body Topology

The design idea of the novel hexagonal cell body topology (NH structure) is shown in Figure 1. The NH structure in this study is based on a single-layer hexagonal cell (Figure 1a), and the overall cell is modified by adding a new hexagon to form the final single-cell microstructure. The single-cell microstructure has two connection points (red points in Figure 1b) for connecting other single-cell microstructures.

The single-cell microstructures are controlled by the following parameters, respectively:*R*: radius of the inner tangent circle of the outer hexagon, which controls the overall size of the microstructures and affects the size of the inner hollow part of the microstructures;*a*: controls the spacing between the inner and outer edge lengths; in this study, this distance is equal to the wall thickness *t*, see Equation (2);*r*: the radius of the inner hexagonal tangent circle of the inner layer, which has the same function as the parameter *R*;*t*: wall thickness.

The relationship of the above parameters is as follows:(1)r=R−a
(2)t=a

Based on the above parameters, the relative density ρ of the NH structure in the face can be expressed as
(3)$$\rho =\frac{\rho_{NH}}{\rho_{s}}=\frac{\left(104R-80t+8\sqrt{3}\pi t\right)t}{3{\left(2R+t\right)}^{2}}$$
where $\rho_{NH}$ is the density of the NH structure and $\rho_{s}$ is the density of the matrix material.

Single-cell microstructures can be rotated and inverted to form multiple forms of microstructures. Multiple single-cell microstructures can be connected through junctions, mimicking the tight arrangement of a natural honeycomb, to form a variety of complex structures. Figure 1b illustrates three complex structures and their basic units, in which all the structures are connected head to tail to form a whole structure. Based on the above connections, many other types of complex structures can also be formed for various scenarios.

### 2.2. Sample Preparation

Both the tensile test samples, according to ISO 527-1:2019 [22], and the impact test samples, according to ISO179-1:2020 [23], were fabricated to investigate the mechanical properties of the NH structure, as well as make a comparison with the properties of the common honeycomb structure. The tensile samples in size are shown in Figure 2a, and the impact samples in size are shown in Figure 2b. Additionally, the impact test samples of the NH structure with single-cell sizes of 1 mm, 1.2 mm, and 1.8 mm were, respectively, made to study the effects of single-cell size on the mechanical properties of the novel structure.

The NH structure (Figure 3a) and the HH structure (Figure 3b) will be filled in the interior of the samples. The thickness of the filler layer for the tensile sample and the impact sample is 0.8 mm and 1.6 mm, respectively. All the test samples in this study were modeled using SolidWorks2022 software. The model files were converted to STL format and then sliced using JGcreat 5.1.0 software to provide the processing information for FDM printing.

### 2.3. Three-Dimensional Printing Process

The physical objects were obtained using an FDM desktop 3D printer produced by Shenzhen Aurora Technology Co., Ltd. (Shenzhen, China). The processed material is polylactic acid filament with a density of 1.21 × 10^3^ kg/m^3^, a diameter of 1.75 mm, and a melting temperature of 190–220 °C. The bulk material has a tensile strength of 62.63 MPa and a bending strength of 65.02 MPa. The parameters set by the 3D printer during the printing process of all the samples are shown in Table 1. Additionally, in the comparison experiments of the two kinds of samples in different cell structures, to ensure the same porosity, the filling rate of the tensile and impact samples filled with the regular honeycomb structure was 74% and 64%, respectively.

### 2.4. Test Methods

Figure 4 illustrates the setup of equipment and samples employed for the tensile test. The test was conducted following the ISO 527-1:2019, utilizing the high-temperature testing machine from Youhong Measurement and Control Technology (Shanghai) Co., Ltd. (Shanghai, China). The machine was configured with a 100 KN tension capacity and a tensile rate of 5 mm/min. Throughout the stretching process, real-time data were processed and recorded by the MDCTest-D 3.2.8 software. The average values of 5 samples were used to evaluate the mechanical properties of the structure.

Figure 5 illustrates the setup of equipment and samples employed for the impact test. The impact test was conducted following the ISO 179-1:2020. The testing was carried out using the impact testing machine for simply supported beam provided by Bangyi Precision Measuring Instrument (Shanghai) Co., Ltd. (Shanghai, China). The pendulum energy of the impact testing machine was set at 5 J, with an impact rate of 2.9 m/s. The average values of 10 samples were used to evaluate the mechanical properties of the structure.

### 2.5. Finite Element Analysis

The numerical model was developed to simulate the in-plane deformation of HH structures and NH structures using hexahedral cells (Solid185). The aim was to investigate the deformation mechanism of the NH structure under uniaxial tension and drop hammer impact, especially the load transfer of the microcells in the NH structure. The geometric parameters of the honeycomb in the numerical model are the same as the experimental samples. The contact in the honeycomb itself is an automatic one-sided contact, and the contact between the honeycomb wall and the outer wall of the model is an automatic face-to-face contact. The sample material is a self-set plastic material, and the material properties come from the data provided by the seller, in which the modulus of elasticity is 2 GPa and the Poisson’s ratio is 0.4. The material of the impact hammer in the impact simulation is the structural steel that comes with the software.

The tensile simulation was analyzed using the hydrostatic module with the boundary condition that the left end face of the specimen was fixed, and a displacement of 5 mm/min was applied to the right end face to simulate the tensile process. The impact simulation analysis was carried out using the LS-DYNA module, and the arrangement is shown in Figure 6. The boundary conditions are fixed on both sides in x-direction, and the long-end side is subjected to impact. Another impact hammer model is added, and the impact hammer impacts the midpoint of the long-end side of the sample with a velocity of 2.9 m/s. The above numerical analysis process was carried out using ANSYS Workbench Student Edition.

A mesh convergence study was conducted on the NH structure to confirm the optimum element size. This analysis examined the force and displacement behavior for different mesh sizes (2.5 mm, 2 mm, 1.5 mm). Prior to the mesh convergence analysis, five standard tensile tests were carried out on the PLA materials used in this study, and stress–strain curves were obtained based on the average of the obtained results (Figure 7a), which were incorporated into the numerical model. The results are shown in Figure 7b, where all three finite element models with different mesh sizes exhibit similar load–displacement behaviors with less differentiation. Considering the computational cost and accuracy, the model with a mesh size of 2 mm was chosen for the subsequent finite element analysis. In addition, since the student version of the software has computational limitations on the number of nodes and cells, all meshes were set as linear cells to further reduce the computational cost.

## 3. Results and Discussion

### 3.1. Experimental Results of Mechanical Properties

#### 3.1.1. Analysis of Tensile Properties of the NH Structure

The results of the tensile experiments conducted on the two structures are illustrated in Figure 8. The stress–strain curves depicting the performance of the HH structure and the NH structure are presented in Figure 8a. The curve results show that both structures are brittle. However, the two structures do not have the same tensile strength and different modulus of elasticity. This proves that the factors affecting the failure mechanism of FDM -printed samples are related to the printing parameters, such as infill pattern and grating angle [24,25]. Different filling patterns caused differences in the tensile strength of the samples. The grating angles of the two samples in this study are angled with respect to the tensile direction, which reduces the elastic modulus of the overall structure [26]. Meanwhile, the results of the elastic modulus are shown in Figure 8b. The elastic modulus of the NH and HH structures decreased by 1.27% and 2.10%, respectively, compared to the solid tensile samples.

The weights of the tensile specimens utilizing the two structures were recorded and are displayed in Figure 8c. The average weight of the HH specimen is 9.85 g, while the NH specimen weighs in at an average of 9.34 g. Comparing the tensile strength of the NH structure to that of the HH structure, it is evident that the former exhibits superior strength, as visually depicted in Figure 8d. The average tensile strength of the HH structure measures 41.14 MPa, whereas that of the NH structure is notably higher at 44.21 MPa.

In summary, the novel hexagonal cell body topology not only boasts a lighter weight profile but also demonstrates a substantial increase in tensile strength when compared to the hexagonal honeycomb structure.

#### 3.1.2. Analysis of Impact Performance of the NH Structure

As shown in Figure 9a, the average weight of the HH impact sample is 3.52 g, while that of the NH impact sample is 3.53 g. The weights of the two structures are almost the same. Subsequently, the impact strength of the two structures is shown in Figure 9b. The impact strength of the HH sample is 9.79 KJ/m^2^, and that of the NH sample is 11.66 KJ/m^2^, representing a 19.1% increase in the impact strength of the NH structure. These results indicate that the mechanical strength of the NH structure is superior as well.

#### 3.1.3. Analysis of Impact Properties for Different Cell Sizes

In this section, the effect of different single-cell size R on the impact performance of the NH was compared. Figure 10 shows the results of impact tests with single-cell sizes of 1 mm, 1.2 mm, and 1.8 mm. From Figure 10a, the average weight of the specimen decreases gradually with the increase in the single-cell size. The impact strength shows a trend of increasing and then decreasing with the increase in single-cell size. Among them, the impact strength of the sample is maximum at a single-cell size of 1.2 mm, which reaches 10.78 KJ/m^2^, and the relative density of the NH structure also reaches its maximum, which is 75.8%; see Figure 10b.

There is a hollow portion inside the novel hexagonal cell body topology, the size of which is determined by the parameter *R* of the single cell. The greater the *R* value caused, the more hollow the portion, which indeed leads to a gradual decrease in the overall weight of the structure. When the single cell size is 1 mm (Figure 11a), the hollow part of the new structure is small, making it closer to the solid structure, but the presence of the tiny hollow becomes a defect in the structure, leading to a decrease in impact strength. As the single cell size increases, making the structure much like the honeycomb structure (Figure 11b), the stress can be uniformly distributed, and the impact strength is improved. However, when the single-cell size is 1.8 mm (Figure 11c), the hollow part increases and the force transfer gradually becomes worse, resulting in a decrease in impact strength instead of an increase. In practical applications, controlling the appropriate single-cell size can improve the impact performance of the NH structure.

### 3.2. Simulation Analysis of the Novel Hexagonal Cell Body Topology

The tensile test results of the HH structure and the NH structure are illustrated in Figure 12a,b. Upon examining the stress distribution in the cloud diagram, it is evident that the highest stress concentration occurs in the narrower section of the samples, ultimately leading to fracture in that region.

The results of the cloud diagrams on the outer surface of the tensile model show that the stress in the narrower part of the HH structure is about 60.5 MPa, while the stress in the NH structure is in the range of 44–52 MPa. This means that the HH structure will reach its fracture strength earlier, and thus, fracture will occur. Figure 12c,d show the results of the cloud maps of the internal structure of the model. They reveal that the microcells in the HH structure experience non-uniform loading, particularly under in-plane tensile forces, where certain cell walls bear insufficient external force. This suggests poor energy absorption efficiency within the microporous structure of the HH structure. Conversely, the NH structure introduced in this study distributes force more evenly across each microcell, enhancing the energy absorption efficiency of the microcell structure significantly. In addition, the results indicate that the single-cell microunits of the NH structure have a stress of 80 MPa at the junction point, which may become a weak point of the overall structure. Further optimization of the microcells for this phenomenon will be carried out in subsequent studies.

Figure 13 displays the simulation results comparing the impact resistance of the HH structure with the NH structure. The findings indicate that at the same impact velocity, the HH structure depicted in Figure 13a starts deforming and fracturing, whereas the model featuring the novel hexagonal cell body topology in Figure 13b remains intact. Upon closer inspection of the magnified images, it is evident that the conventional honeycomb structure in Figure 13c fails under the impact of the outer layer, with individual units struggling to absorb the impact, resulting in a compromised overall impact resistance. In contrast, each unit of the novel hexagonal cell body topology in Figure 13d effectively absorbs and transfers the impact to adjacent units, showcasing superior impact resistance capabilities. This highlights the enhanced impact resistance of the novel hexagonal cell body topology structure compared to the traditional honeycomb design.

## 4. Conclusions

Drawing inspiration from the natural honeycomb structures, a novel hexagonal cell body topology was introduced. Comparative experiments on the tensile and impact properties of the new structure and the regular honeycomb structure were carried out, as well as comparative experiments on the impact properties of the new structure with different single cell sizes. Furthermore, finite element simulations were employed to scrutinize the forces acting on the microscopic cells within the new structure. The key findings are as follows:In the structural comparison experiment, the tensile strength of the novel hexagonal cell body topology reaches 44.21 MPa, representing a 7.46% increase compared to the conventional honeycomb structure. And the impact strength of the novel hexagonal cell body topology measures 11.66 KJ/m^2^, marking a substantial 19.1% improvement over the ordinary honeycomb structure.The size of the single cell directly affects the size of the hollow part inside the structure, which influences the impact performance of the overall structure, and if the size of the single cell is too large or too small, it will reduce the impact performance of the overall structure. It was found that the impact properties of the specimens were optimal at a single cell size of about 1.2 mm. How to determine the optimal single-cell size remains to be investigated.Finite element analysis reveals that each element within the novel hexagonal cell body topology excels in transferring external forces, leading to a more uniform stress distribution and enhanced tensile and impact strength of the sample.

## Figures and Tables

**Figure 1 polymers-16-02201-f001:**
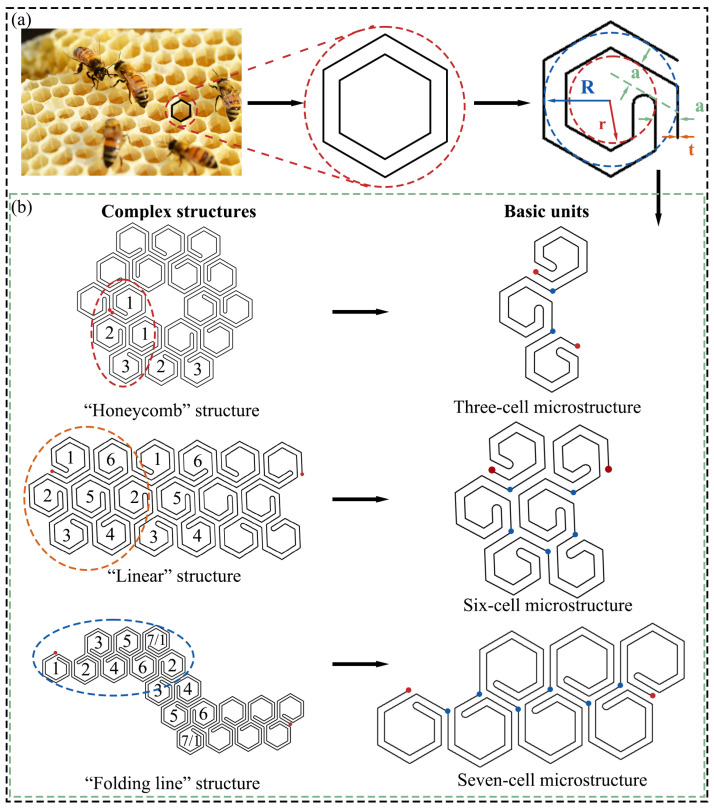
Schematic diagram of the novel hexagonal cell body topology. (**a**) Evolution of the microstructure of single-cell body. (**b**) Multiple derived structures of the NH structure.

**Figure 2 polymers-16-02201-f002:**
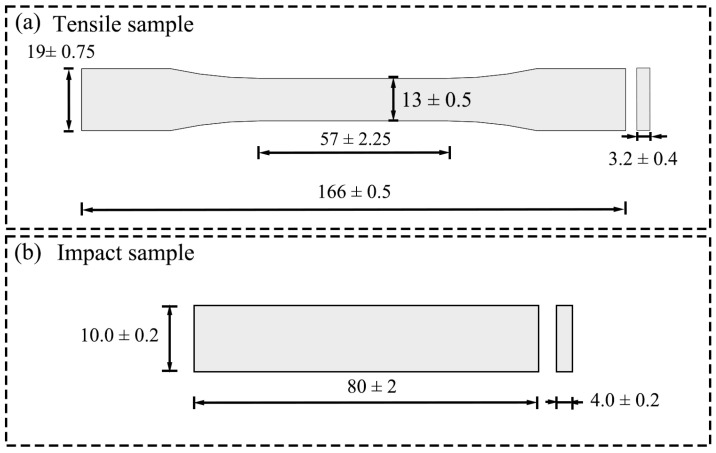
Dimensions of tensile and impact specimens. (**a**) Tensile sample. (**b**) Impact sample.

**Figure 3 polymers-16-02201-f003:**
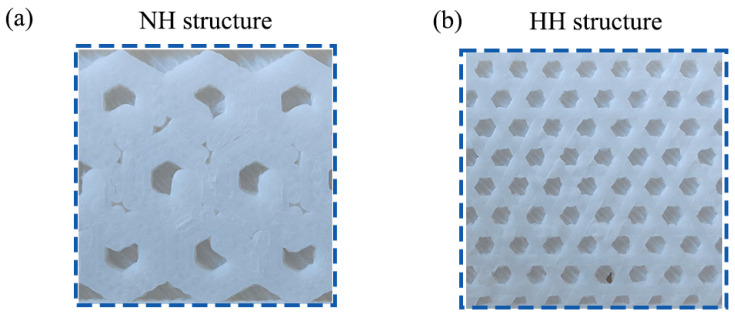
Schematic diagram of the filler layer inside the printed samples. (**a**) NH structure. (**b**) HH structure.

**Figure 4 polymers-16-02201-f004:**
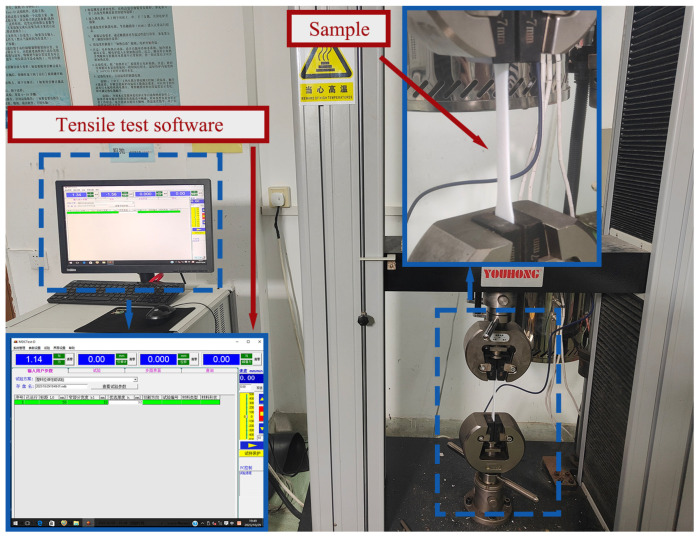
Process of tensile test.

**Figure 5 polymers-16-02201-f005:**
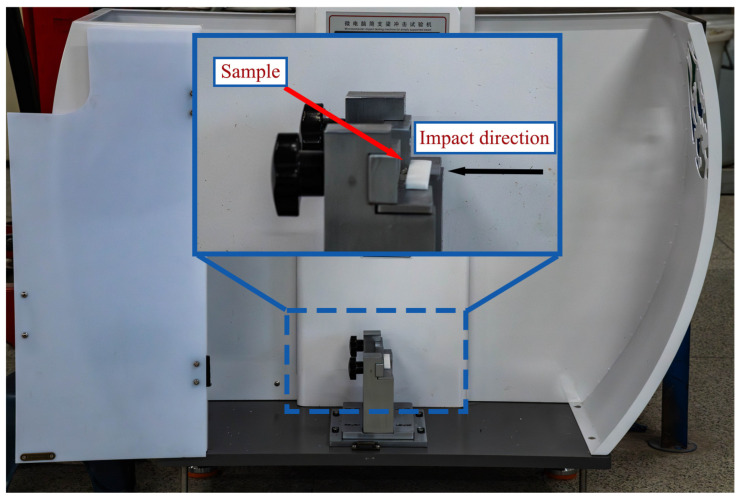
Process of impact test.

**Figure 6 polymers-16-02201-f006:**
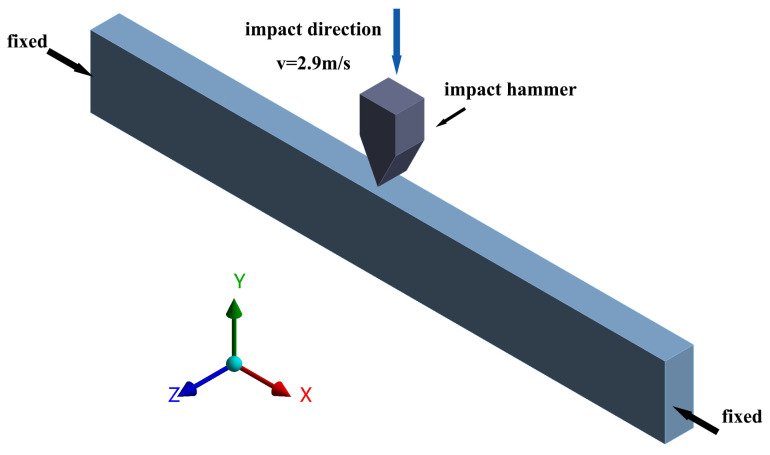
Finite element modeling of impact sample.

**Figure 7 polymers-16-02201-f007:**
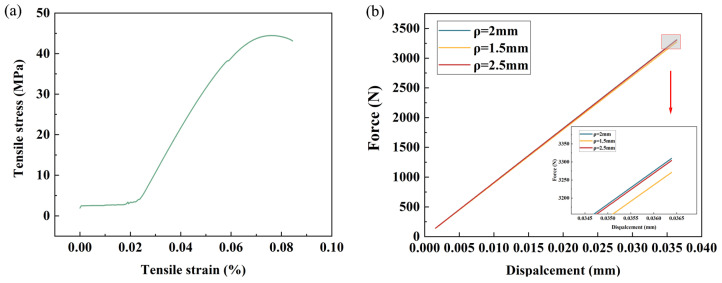
Mesh convergence study. (**a**) Stress–strain curve of the PLA material. (**b**) Simulation results of three different mesh sizes.

**Figure 8 polymers-16-02201-f008:**
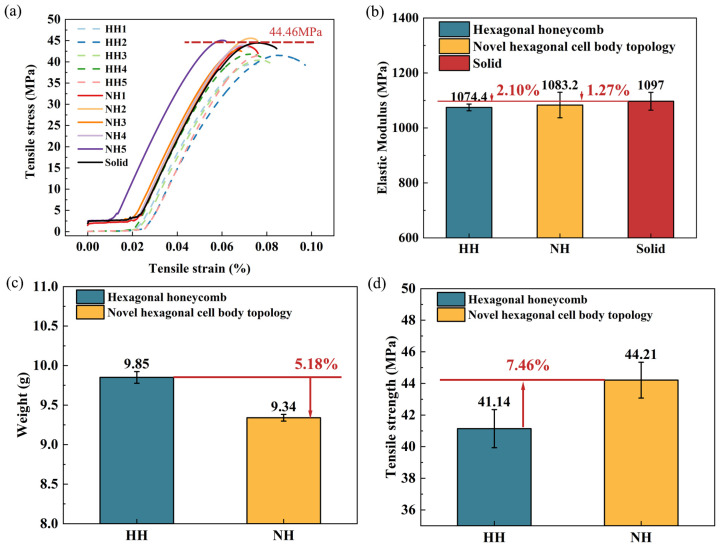
Tensile test results of hexagonal honeycomb structure and novel hexagonal cell body topology. (**a**) Stress–strain curves. (**b**) Elastic Modulus. (**c**) Tensile sample weight. (**d**) Tensile strength.

**Figure 9 polymers-16-02201-f009:**
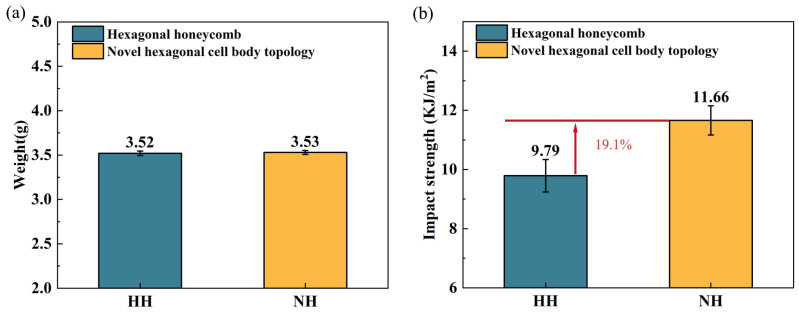
Impact test results of hexagonal honeycomb structure and novel hexagonal cell body topology. (**a**) Impact sample weight. (**b**) Impact strength.

**Figure 10 polymers-16-02201-f010:**
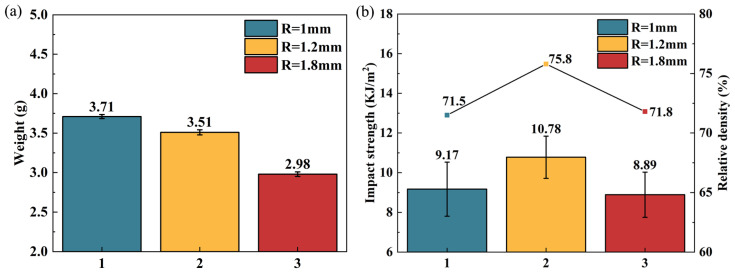
Impact results of samples with different cell sizes. (**a**) Impact sample weight. (**b**) Impact strength.

**Figure 11 polymers-16-02201-f011:**
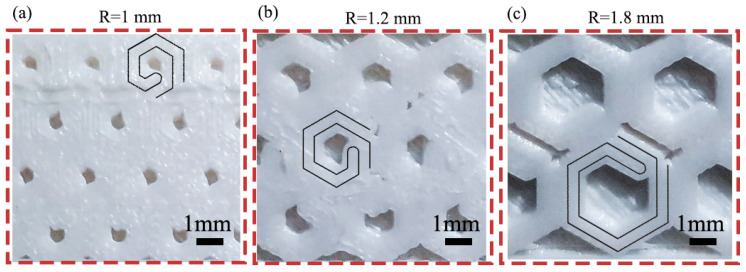
Cell body with the R equal to (**a**) 1 mm, (**b**) 1.2 mm, (**c**) 1.8 mm.

**Figure 12 polymers-16-02201-f012:**
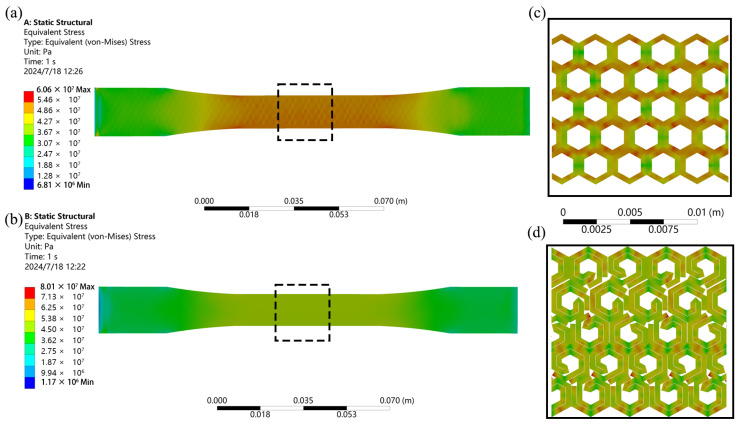
Simulation results of tensile model. (**a**) Contour plot of the overall HH structure. (**b**) Contour plot of the overall NH structure. (**c**) Localized enlarged view of the HH structure. (**d**) Localized enlarged view of the NH structure.

**Figure 13 polymers-16-02201-f013:**
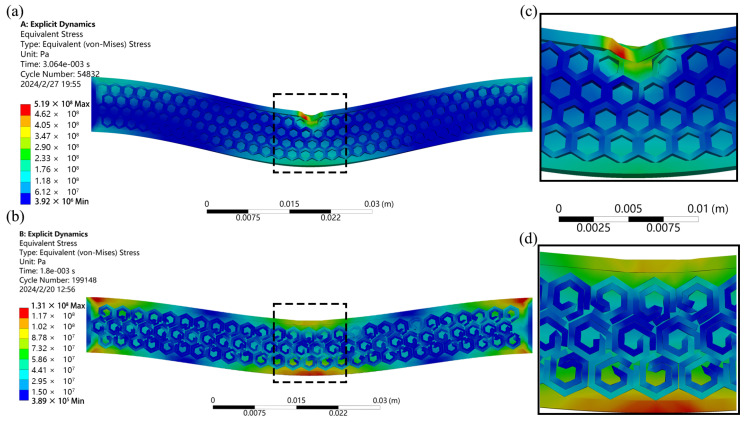
Simulation results of impact model. (**a**) Contour plot of the overall HH structure. (**b**) Contour plot of the overall NH structure. (**c**) Localized enlarged view of the HH structure. (**d**) Localized enlarged view of the NH structure.

**Table 1 polymers-16-02201-t001:** Parameters of the 3D printer.

Nozzle Diameter (mm)	Printing Speed (mm/s)	Printing Temperature (°C)	Build Plate Temperature (°C)	Layer Thickness (mm)	Top/Bottom Thickness (mm)
0.4	50	210	50	0.1	1.2

## Data Availability

The original contributions presented in the study are included in the article, further inquiries can be directed to the corresponding author.

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
