# Peer review of "Study on the Design and Mechanical Properties of a Novel Hexagonal Cell Body Topology"

_polymers, 2024, doi:10.3390/polym16152201_

Round 1
Reviewer 1 Report
Comments and Suggestions for Authors
In this research, inspired by natural honeycomb structures, a new hexagonal cell body topology was introduced and manufactured by the (FDM) technique. Comparative tests were performed on the tensile and impact properties of the new structure and the conventional honeycomb structure and different sizes of single cells. In addition, finite element simulations were used to accurately investigate the forces acting on the microscopic cells in the new structure. In the structural comparison test, the tensile strength of the new hexagonal cellular body topology reaches 44.21MPa, which represents a 7.46% increase over the conventional honeycomb structure. But excessively large or small single cell sizes of the NH structure reduce the overall impact resistance of the structure and The overall structure achieves optimal impact resistance when the single cell size is around 1.2 mm.
· Did the authors adequately explain the observed trends in stress-strain curves for both structures? Is the brittleness observed a limitation of the PLA material or the manufacturing process?
· Use the following references. 3D printing of PLA-TPU with different component ratios: Fracture toughness, mechanical properties, and morphology. Experimental investigation on mechanical characterization of 3D printed PLA produced by fused deposition modeling (FDM).
· Review figure captions. Please complete the description of figure1.
· The text mentions "tightly arranging multiple monocytes connected through their respective junctions." Could this be elaborated on? Are there different connection methods or configurations possible?
· Is the function of parameter R (radius of inner tangent circle) adequately explained? How does it affect the size and potentially other properties of the single cells?
Reviewer 2 Report
Comments and Suggestions for Authors
Dear Author,
I have read the manuscript titled "Study on the design and mechanical properties of a novel hexagonal cell body topology," which explores the honeycomb structure's topological advantages stemming from its fundamental units. This study designed and manufactured a novel hexagonal cell body topology (NH) using the fused deposition modelling (FDM) technique to investigate its mechanical properties. The tensile and impact performance of the NH structure was compared with the regular hexagonal honeycomb structure (HH), and the impact of varying single-cell sizes on the NH structure's performance was assessed. Finite element analysis revealed improved force transmission in the NH structure, leading to enhanced tensile and impact performance due to more uniform stress distribution.
However, the paper is largely conceptual, with the finite element analysis lacking proper verification, rendering the conclusions rudimentary. Therefore, I suggest Major Revisions, with increased focus on analysis, verification of computational models, detailed model descriptions, and specifically on the impact model and its results.
The following are some comments for the authors:
Line 36-44: This part does not add any significant value to the introduction or literature review. It is recommended to either rewrite it to better integrate with the core objectives and context of the study or exclude it from the text altogether.
Figure 1: The title for part "b" in Figure 1 is missing. Please add an appropriate title to ensure clarity and completeness of the figure.
Section 2.4: Adding a picture of the impact test setup would significantly enhance the clarity and value of the test description.
Section 2.5: The finite element (FE) model used for structural components needs a detailed description, including the type of finite element itself, such as degrees of freedom, possible deformations, and other relevant aspects. The material model should be thoroughly introduced. Additionally, it is essential to explain how the topology of the structures was created, whether there were any interlinks between layers, the number of FEs used per structural element thickness, and specifically how the setup for simulating the impact test was created.
Figure 5: In Figure 5a, it is unclear which type of material the stress-strain curve represents. Additionally, only one curve is presented. Is this curve an average result? Please provide clarification and, if applicable, include multiple curves to show variability or specify that the presented curve is an average.
Figure 6: Why are the results of NH5 so different from the others in terms of stiffness? The stiffness comparison is missing, and the percentage differences in stiffness are not provided. It is recommended to compare both materials with the results of bulk material and provide a proper quantified analysis.
Line 247: The graphical results in Figure 10 show the maximum stress for NH is 80 MPa, which contradicts the statement in the text. Please clarify this discrepancy.
Line 253-255: How can you determine from the stress diagram that some regions reach material strength prematurely? This statement needs clarification. Please provide additional details or evidence to support this claim.
Line 267: Instead of "initial," the more appropriate word is "outer."
Reviewer 3 Report
Comments and Suggestions for Authors
the research domain is not new but is still under consideration
the proposed idea concerns new biomimetic structures derived from honeycomb
at least a minimal mathematic model of existing and new proposed structure must be provided
i suggest the authors to consult
https://doi.org/10.1016/j.ijimpeng.2023.104641, https://doi.org/10.1016/j.ijmecsci.2022.107718 and https://doi.org/10.1016/j.jmrt.2022.12.063
the conclusions are in accordance with the presented experiments over proposed structure
Round 2
Reviewer 1 Report
Comments and Suggestions for Authors
Accept.